# Protein Intake at Twice the RDA in Older Men Increases Circulatory Concentrations of the Microbiome Metabolite Trimethylamine-N-Oxide (TMAO)

**DOI:** 10.3390/nu11092207

**Published:** 2019-09-12

**Authors:** Sarah M. Mitchell, Amber M. Milan, Cameron J. Mitchell, Nicola A. Gillies, Randall F. D’Souza, Nina Zeng, Farha Ramzan, Pankaja Sharma, Scott O. Knowles, Nicole C. Roy, Anders Sjödin, Karl-Heinz Wagner, Steven H. Zeisel, David Cameron-Smith

**Affiliations:** 1Liggins Institute, The University of Auckland, Auckland 1023, New Zealand; sarah.mitchell@auckland.ac.nz (S.M.M.); a.milan@auckland.ac.nz (A.M.M.); cameron.mitchell@ubc.ca (C.J.M.); n.gillies@auckland.ac.nz (N.A.G.); r.dsouza@auckland.ac.nz (R.F.D.); n.zeng@auckland.ac.nz (N.Z.); f.ramzan@auckland.ac.nz (F.R.); p.sharma@auckland.ac.nz (P.S.); 2Food Nutrition & Health Team, Food & Bio-based Products Group, AgResearch, Palmerston North 4442, New Zealand; scott.knowles@agresearch.co.nz (S.O.K.); nicole.roy@agresearch.co.nz (N.C.R.); 3School of Kinesiology, University of British Columbia, Vancouver, BC V6T 1Z1, Canada; 4Discipline of Nutrition, School of Medical Sciences, University of Auckland, Auckland 1023, New Zealand; 5Riddet Institute, Massey University, Palmerston North 4442, New Zealand; 6The High-Value Nutrition National Science Challenge, Auckland 1023, New Zealand; 7Department of Nutrition, Exercise and Sport, Copenhagen University, 1165 Copenhagen, Denmark; amsj@nexs.ku.dk; 8Department of Nutritional Sciences and Research Platform Active Ageing, University of Vienna, 1010 Vienna, Austria; karl-heinz.wagner@univie.ac.at; 9Nutrition Research Institute, University of North Carolina, Kannapolis, NC 28081, USA; steven_zeisel@unc.edu; 10Clinical Nutrition Research Centre, Singapore Institute for Clinical Sciences, Agency for Science, Technology and Research, Singapore 117609, Singapore

**Keywords:** TMAO, high protein diet, CVD, CVD biomarkers, elderly

## Abstract

Higher dietary protein intake is increasingly recommended for the elderly; however, high protein diets have also been linked to increased cardiovascular disease (CVD) risk. Trimethylamine-N-oxide (TMAO) is a bacterial metabolite derived from choline and carnitine abundant from animal protein-rich foods. TMAO may be a novel biomarker for heightened CVD risk. The purpose of this study was to assess the impact of a high protein diet on TMAO. Healthy men (74.2 ± 3.6 years, n = 29) were randomised to consume the recommended dietary allowance of protein (RDA: 0.8 g protein/kg bodyweight/day) or twice the RDA (2RDA) as part of a supplied diet for 10 weeks. Fasting blood samples were collected pre- and post-intervention for measurement of TMAO, blood lipids, glucose tolerance, insulin sensitivity, and inflammatory biomarkers. An oral glucose tolerance test was also performed. In comparison with RDA, the 2RDA diet increased circulatory TMAO (*p* = 0.002) but unexpectedly decreased renal excretion of TMAO (*p* = 0.003). LDL cholesterol was increased in 2RDA compared to RDA (*p* = 0.049), but no differences in other biomarkers of CVD risk and insulin sensitivity were evident between groups. In conclusion, circulatory TMAO is responsive to changes in dietary protein intake in older healthy males.

## 1. Introduction

Adequate protein intake is key for the elderly [1]. The current recommendations for protein intake for all adults, according to the World Health Organization (WHO) [2] and the US Department of Agriculture (USDA) [3], is 0.8 g protein/kg body weight/day. However, a growing number of randomised controlled trials (RCTs) [4], mechanistic [5] and epidemiological [6] studies support an increase from the current recommendations to mitigate the risks of age-related loss of skeletal muscle (sarcopenia), falls, and associated mortality risk [7,8,9]. Yet, there is also the possibility that a high protein diet may increase the risk for cardiovascular disease (CVD). Epidemiological evidence suggests that high protein diets from both plant and animal sources are associated with reduced CVD risk [10,11]. Furthermore, interventional studies have demonstrated that during an energy deficit, a higher protein intake might be beneficial for cardiovascular and metabolic health, particularly in obese and at-risk populations [12,13,14,15]. Conversely, further evidence from epidemiological and association studies suggests that high protein intake heightens the risk of CVD [16,17,18,19]. High protein diets are also associated with an increased prevalence of diabetes [20,21] and impaired kidney function [22]. 

A putative biomarker for CVD risk is the gut microbiota-derived metabolite trimethylamine-N-oxide (TMAO) as it has repeatedly been associated with an increased risk of adverse cardiovascular events [23,24,25,26]. Dietary intake of the nutrients choline and carnitine are the main contributors to circulatory TMAO via their metabolism by gut microbiota. Choline is an essential nutrient abundant in protein-rich foods, such as eggs, red meat, poultry, and fish [27]; similarly, carnitine is found at high levels in red meat, poultry, and dairy products [28]. Clinical studies have demonstrated that plasma and urine TMAO concentrations increase transiently after ingestion of choline or carnitine supplements [29], red meat [25], fish [30], dairy [31], and egg ingestion [32]. 

There are limited RCTs that have analysed the impact of high protein dietary interventions on circulatory TMAO concentrations, particularly for the elderly. Consequently, it is inconclusive whether high protein diets modify fasting plasma TMAO concentrations in this population. Hence, the purpose of this study was to determine the effect of the consumption of dietary protein at twice the current RDA compared to consumption at the RDA on TMAO concentrations. Further analysis was made of traditional biomarkers of cardiovascular and metabolic disease risk in the elderly. 

## 2. Materials and Methods 

### 2.1. Study Design

The study was a 10-week single-blind parallel randomised controlled trial. The study was approved by the Southern Health and Disability Ethics Committee (New Zealand; 15/STH/236) and was conducted in accordance with the Declaration of Helsinki. The study protocol was registered with the Australia New Zealand Clinical Trial Registry #ACTRN12616000310460. All the subjects gave written informed consent to participate in the study. 

### 2.2. Participants

Thirty-one men were recruited to take part in the study (Table 1) via public advertisements and were screened by questionnaire. Eligible subjects were males aged 70 years or more, non-smokers, had a body mass index (BMI) between 20 and 32 kg/m^2^, and were independent community dwellers. The exclusion criteria included prior history of cardiovascular, metabolic, renal, or gastrointestinal disease. Participants were recruited from the Auckland (New Zealand) region and data was collected at the Liggins Institute, University of Auckland, between March and October 2016.

### 2.3. Study Procedures

Participants were randomly assigned to a dietary intervention for 10 weeks according to a parallel design: (1) the current recommended protein diet (RDA group; 0.8 g protein/kg bodyweight/day), or (2) a high protein diet (2RDA group; twice the RDA). This study reports secondary outcomes from a clinical trial for which the primary outcomes were changes in muscle size and strength [4]. The quantity of protein in the 2RDA diet was chosen as it has previously been recommended as optimal for preserving muscle mass in older adults [33]. 

Sample size calculation, randomisation, dietary and compliance data was described previously [4]. Briefly, both diets contained approximately 30% energy from fat with the balance made up of carbohydrates. Protein was made up from a combination of animal and plant sources, including dairy, eggs, poultry, fish, red meat, legumes, grains, nuts, and seeds. All food consumed was provided and delivered weekly to the participants’ homes. The energy contents of the diets were individually calculated based on the Harris-Benedict equation [34] and adjusted for physical activity as assessed by a wrist accelerometer (Fitbit Charge HR; Fitbit, Inc., San Francisco, CA, USA). During the intervention, participants attended the Nutrition Clinic at the Liggins Institute at weeks 4, 6 and 8 for weight measurement and compliance check. Participants were asked to make no changes to their exercise routine throughout their participation in the trial. Fasting anthropometry measurements and biological sampling, and an oral glucose tolerance test (OGTT) were taken at pre-intervention (Pre) and after 10 weeks of intervention (Post). 

### 2.4. Dietary Analysis

Prior to the study, participants completed a 3-day food diary at baseline to determine habitual intake and completed fortnightly checklists throughout the study to monitor assigned dietary intake. Macronutrient intakes were estimated by a registered dietician using Foodworks software (Version 8; Xyris). Serving sizes of eggs (1 egg), fish (100 g), meat (100 g), and dairy (250 mL milk, 50 g cheese, 150 mL yoghurt) were calculated from the habitual diet and study diets, as these were collectively the major contributors of choline and carnitine in the diet. 

### 2.5. Sample Collection and Analysis

Fasting blood samples were drawn from an antecubital vein by cannula into both EDTA-coated vacutainers and uncoated serum vacutainers. Serum was left to clot for 20 min before all the tubes were centrifuged for 15 min at 1900× *g* and 4 °C. Plasma and serum samples were then stored at −80 °C until analysis.

TMAO, choline, and betaine were quantified in plasma and urine using stable isotope dilution coupled to liquid chromatography-multiple reaction monitoring mass-spectrometry with an ACQUITY UPLC (Waters, Milford, MA, USA), according to previously published methods [35].

Carnitine in plasma was quantified using reversed-phase ultra-high-performance liquid chromatography coupled with tandem mass-spectrometry (UHPLC-MS/MS) (ThermoFisher Scientific, Santa Clara, CA, USA).

Plasma was measured for glucose, creatinine, and lipids (including total cholesterol (T-C), low-density lipoprotein cholesterol (LDL-C), high-density lipoprotein cholesterol (HDL-C), and triacylglycerides (TAG)), and serum was assessed for C-reactive protein (CRP) by a Cobas C311 Analyser (Roche Diagnostics, Basel, Switzerland) by enzymatic colorimetric assays. Insulin concentration in plasma was measured by a Cobas e411 (Roche Diagnostics, Basel, Switzerland) by immunoassays. Whole blood was measured for glycated hemoglobin (HbA1c) using the Cobas C311 Analyser (Roche Diagnostics, Basal, Switzerland). 

Participants were asked pre- and post-intervention to collect a 24-hour urine sample in a container provided. Samples were weighed and stored in aliquots at −80 °C. The concentration of urinary nitrogen (comprising urea, creatinine, uric acid) was measured with a Cobas C311 Analyser (Roche Diagnostics, Basel, Switzerland) to estimate dietary protein intake and nitrogen balance [36]. 

### 2.6. OGTT

Fasting sample collection was followed by a 2-h OGTT whereby a pre-prepared glucose solution containing 75 g of glucose as a 200-mL beverage was consumed within 5 min. During the OGTT, blood samples were collected in 30-min intervals for 2 h (0, 30, 60, 90, and 120 min) for the measurement of plasma glucose and insulin concentrations to establish whole-body insulin sensitivity, calculated by the Matsuda Index [37] and by using the homeostatic model assessment of insulin resistance (HOMA-IR) [38].

### 2.7. Anthropometry

Resting blood pressure was measured with an electronic sphygmomanometer (Omron HEM-71111-AU, Omron Healthcare, Kyoto, Japan). Participants were weighed with a calibrated scale (Tanita HD-351, Arlington Heights, IL, USA) without shoes and height was measured with a stadiometer (HoltainLtd., Crymych, UK). BMI was calculated using these data. Waist circumference was taken with a tape measure at the top of the iliac crest. All measurements were taken twice and the average was reported. Dual-energy X-ray absorptiometry (DXA) was performed for total fat mass and central adiposity (android % fat) (Prodigy, GE-Lunar, Madison, WI, USA). 

### 2.8. Calculations

Creatinine clearance was calculated from plasma and 24-h urine creatinine excretion as follows: 

Creatinine clearance = [(urine creatinine concentration in mmol/L) × (urine volume in mL/1140 min)/(plasma creatinine concentration in (μmol/L) · 1000 mL^−1^ · min^−1^)] [39]. 

Fractional renal excretion of TMAO was calculated as follows: 

Fractional renal excretion of TMAO = (urinary TMAO × plasma creatinine)/(plasma TMAO × urinary creatinine) × 100 [40].

### 2.9. Statistical Analysis

Sample size was calculated for the primary outcomes which were changes in muscle size and strength [4]. Data are presented as mean ± standard deviation (SD). Statistical analyses were performed using the software package SPSS, version 25.0 (SPSS Inc., Chicago, IL, USA). Between-group differences were analysed using a two-way repeated measures ANOVA with time (from pre to post) as a repeated factor and diet (RDA vs 2RDA) as a between-subject factor. Post hoc multiple comparisons were controlled using Sidak corrections. Normality was assessed with the Shapiro–Wilk test and non-normally distributed data were log transformed before further analysis. Correlation analyses between variables were determined using Spearman’s correlation coefficient for non-normally distributed data. Significance was accepted at *p* < 0.05. Unadjusted means ± SDs are shown in the tables and text. 

## 3. Results

Thirty-one participants were randomly assigned to the two diets (RDA: n = 16, 2RDA: n = 15). One withdrew before the start of the intervention (RDA) and one was excluded due to non-compliance (2RDA). The pre-intervention characteristics of the remaining 29 participants are summarised in Table 1 and reported in detail elsewhere [4]. 

### 3.1. Dietary Intake

A comprehensive summary of macronutrient intake has been published [4]. Overall, participants were highly compliant to both diets (compliance for protein intake was 98.9% in the RDA group and 97.5% in the 2RDA group). This was documented in fortnightly compliance food records and according to urinary nitrogen excretion assessed pre- and post-intervention (RDA: from 1.3 ± 0.2 to 0.9 ± 0.2 g/kg bodyweight/day, *p* = 0.001; 2RDA: from 1.3 ± 0.2 to 1.5 ± 0.1 g/kg bodyweight/day, *p* = 0.001). Average protein intake was altered by the intervention (time × diet interaction *p* < 0.001); it reduced in the RDA group (from 104.8 ± 30 to 80.1 ± 23 g/day, *p* = 0.004) and increased in the 2RDA group (from 95.6 ± 20 to 136.2 ± 18 g/day, *p* < 0.001). Dietary cholesterol decreased in the RDA group (from 386.8 ± 142.1 to 299.2 ± 83.2 mg/day, *p* = 0.063) and increased in the 2RDA group (from 351.1 ± 113.4 to 526.8 ± 73.4 mg/day, *p* < 0.001) as a result of the intervention (time × diet interaction *p* < 0.001). Dietary fibre increased (time effect, *p* < 0.001) for RDA (from 34 ± 9.9 to 47 ± 8.4 g/day, *p* < 0.001), and for 2RDA (from 28.9 ± 8.1 to 50.3 ± 5.3 g/day, *p* < 0.001) but was not different between the diets (*p* = 0.410) as a consequence of including the local recommendations for fruit and vegetable intake in both diets [41]. Egg, fish, red meat, white meat, and dairy which were the major contributors to choline and carnitine intake are listed in Table 2. 

### 3.2. TMAO, Choline, Betaine, and Carnitine 

The concentrations of TMAO, choline, betaine, and carnitine in plasma are shown in Figure 1, and the concentrations of TMAO, choline, and betaine in urine (expressed as the ratio to urinary creatinine concentrations) are shown in Figure 2. There was an interaction effect of plasma TMAO (time × diet interaction, *p* = 0.002; Figure 1A) which did not change in the RDA group (12.8 ± 9.67 µM vs. 8.05 ± 7.52 µM, *p* = 0.165) and increased in the 2RDA group (from 8.34 ± 4.79 µM to 29.08 ± 31.53 µM, *p* = 0.004). There was an interaction effect of plasma betaine (time × diet interaction *p* = 0.006; Figure 1C) which increased in the RDA group (from 38.67 ± 5.25 µM to 43.73 ± 9.16 µM, *p* = 0.018) and was unchanged in the 2RDA group (from 46.71 ± 13.26 µM to 44.07 ± 11.18 µM, *p* = 0.219). However, there was a significant difference between groups pre- but not post-intervention (*p* = 0.030, *p* = 0.930, respectively) which may have driven this effect. There was no effect of the intervention on plasma choline or carnitine (time × diet interaction *p* = 0.982 and *p* = 0.492, respectively). There was an interaction effect of TMAO in urine (time × diet interaction, *p* = 0.007) which did not change in the RDA group (77.4 ± 77 vs. 125.2 ± 134 mmol/mol creatinine, *p* = 0.345) and decreased in the 2RDA group (from 209.8 ± 198 to 77.7 ± 46.2 mmol/mol creatinine, *p* = 0.005; Figure 2A). There was no interaction effect in urinary choline or betaine (time × diet interaction *p* = 0.389 and *p* = 0.656, respectively) although urinary betaine decreased in both groups (time effect, *p* = 0.032). Change in plasma TMAO correlated inversely with change in urinary TMAO (r = −0.698, *p* = 0.01). 

### 3.3. Lipids and Lipoproteins

The effects on plasma lipids and lipoproteins are shown in Table 3. Small changes in LDL-C concentrations were dependent on diet and decreased in the RDA group and increased in the 2RDA group (time × diet interaction, *p* = 0.049). There were no significant changes in T-C, HDL-C or TAG in either group. 

### 3.4. Metabolic and Inflammatory Biomarkers

Effects on metabolic and inflammatory biomarkers are shown in Table 3. There was a decrease in fasting blood glucose in both groups (time effect, *p* = 0.020) without interaction. Other biomarkers of glucose tolerance and insulin sensitivity, namely insulin, HOMA-IR, and HbA1c, were not affected by the intervention in either group, nor were the results of the OGTT. CRP concentrations were also not affected by the dietary intervention.

### 3.5. Body Composition and Blood Pressure

The effects of selected measures of body composition are summarised in Table 3 and changes in lean mass and total body weight are reported in detail elsewhere [4]. There was an effect of the intervention on android (abdominal fat deposition) % fat (time × diet interaction, *p* = 0.049; Figure 3). Total fat mass and waist circumference also decreased in both groups (time effect, all *p* < 0.001). There was no effect of time or diet on blood pressure.

### 3.6. Creatinine and Creatinine Clearance

The effects on creatinine in plasma and urine and creatinine clearance are summarised in Table 4. There was an interaction effect on plasma creatinine concentrations (time × diet interaction, *p* = 0.020), which decreased in the RDA group (*p* = 0.014) but was unchanged in the 2RDA group (*p* = 0.776). This contributed to a significant difference in creatinine clearance (time × diet interaction, *p* = 0.013) whereby it increased in the RDA group (*p* = 0.010) with no change in the 2RDA group. There was an interaction effect of the fraction renal excretion of TMAO (time × diet interaction, *p* = 0.002) which increased insignificantly in the RDA group (*p* = 0.229) and decreased in the 2RDA group (*p* = 0.002).

## 4. Discussion

This study demonstrates that in the short term (10 weeks), a diet containing twice the RDA of protein can elevate fasting circulatory TMAO. Furthermore, this dietary interventional also resulted in a marginal increase in LDL-C. However, in contrast to the hypothesis, a high protein diet does not affect other established biomarkers of CVD risk and insulin sensitivity. 

After 10 weeks of the 2RDA diet, an increase in plasma TMAO concentrations was observed in the majority of these participants, where on average, plasma TMAO increased five-fold. This response follows previously reported studies where the dietary precursors of TMAO were increased. For instance, observational studies demonstrated a positive correlation between fasting TMAO plasma concentrations and self-reported red meat and fish intake [30] and comparatively higher concentrations of TMAO were found in the plasma and urine of omnivores compared to vegetarians [24,42]. Randomised studies showed that chronic ingestion of red meat for 4 weeks [40] and daily carnitine supplementation for 12 weeks [43] elevated fasting TMAO plasma concentrations by three-fold and ten-fold, respectively. In acute dietary trials, ingestion of meats, fish, eggs, and dairy resulted in postprandial transient increases in TMAO in the circulation and in urine, peaking at around 6–8 hours [32], or up to 12 hours after eating fish [44].

The TMAO concentrations pre-intervention were diverse and the magnitude of response was varied, particularly in the 2RDA group. In the present study, the pre-intervention TMAO values averaged 10 µM, which is higher than the reported reference values of circulatory TMAO in healthy adults, which are in the range of 3.5 µM [45]. However, previous studies have reported that TMAO concentrations increase with age [45,46]. The inter-individual variation of circulatory TMAO is not unique to this study and can be driven by several factors. In addition to food intake, the gut microbiota strongly influences circulatory concentrations of TMAO [47]. A higher Firmicutes to Bacteroidetes ratio and reduced microbial diversity is related to a greater TMAO production [29]. TMAO is not only produced by microbiota fermentation of dietary choline and carnitine from TMA [24,25] but can also be absorbed directly from eating fish [29]. Indeed, fish is the highest source of circulatory TMAO in postprandial studies when compared to other foods enriched in choline and carnitine [29]. The conversion of TMA produced by the gut microbiota to TMAO is dependent on hepatic flavin-containing monooxygenases (FMOs) [48], therefore, variations in the expression of hepatic FMO can affect TMAO concentrations. Additionally, as the majority of TMAO is filtered through the kidneys and excreted through the urine [49], renal function can have an impact on circulatory TMAO. In our study, the concentrations of circulatory TMAO in the 2RDA group post-intervention reached comparable levels described in a previous carnitine supplementation study [43] although in some, the concentrations of TMAO reached levels seen in chronic kidney disease (CKD) patients, which can be 40-fold greater relative to healthy cohorts [50].

Renal function calculated from creatinine clearance was within the reported reference values for healthy males, which are between 97 to 137 mL/min [51]. The 2RDA diet did not have an effect on creatinine clearance, which is in agreement with findings from a previous study in older pre-diabetic adults [52], although the effect of higher protein intakes on renal function in healthy adults continues to be debated [53]. Kidney function is an independent risk factor for CVD [54], and impaired renal function is correlated with elevated plasma TMAO concentrations [55,56]. Although creatinine clearance was not affected in the 2RDA group, urinary TMAO levels and the fractional rate of TMAO clearance decreased. This is interesting as it was expected that urinary TMAO concentrations would increase correspondingly with circulating TMAO, such as has been observed in previous dietary studies with high dietary TMAO precursors [29,40]. In particular, the change in urinary TMAO concentrations in the current study was inversely correlated with plasma TMAO concentrations. This suggests that reduced efficiency in renal clearance for TMAO may be a contributory factor alongside increased protein intake for elevated circulatory TMAO.

Despite the observed increases in TMAO, these did not correspond with concurrent alterations in CVD biomarkers within a 10-week period, although there was an increase in LDL-C in the 2RDA group. Circulating lipids are a common proxy for the prediction of CVD risk and elevated circulating levels of LDL-C and in particular, have a pivotal role in the initiation and progression of atherosclerosis [57]. Despite this, HDL-C, T-C, and TAGs were unchanged by the dietary intervention. The increased LDL-C is in contrast with previous studies in overweight adults [58] and type 2 diabetics [59] where there was no effect of increased protein intake on LDL-C, and a recent meta-analysis reported no effect of higher protein diets on LDL-C [60]. In the present study, dietary cholesterol intake also increased in the 2RDA group from pre-intervention compared to the RDA group. Although still controversial, a recent meta-analysis indicated that while dietary cholesterol may not be associated with CVD, there is evidence that it does increase both total cholesterol and LDL-C [61], which could be a contributory factor toward the observed increase in LDL-C.

There were improvements in some additional indicators of metabolic health observed in both groups that were not unique to the 2RDA diet. Central adiposity (measured by android % fat and waist circumference) decreased in both groups. Numerous short-term high protein diet studies have demonstrated increased weight loss [62] and reduction in other indices, such as waist circumference [63,64], however, these studies are often conducted in populations who are obese or with metabolic disease. Since the fat loss was also observed in the RDA group, it is possible that the increased fibre intake in both groups contributed to the overall observed fat loss [65]. Fasting insulin and insulin sensitivity (calculated by the Matsuda Index [37]) was not changed in either group, and fasting glucose decreased in both groups over time.

A strength of this study was that the macronutrient components were entirely composed of wholefoods rather than relying on supplements. However, certain dietary components may also have had an inadvertent influence on the measured outcomes. For example, circulatory TMAO levels were elevated significantly by seafood intake for up to 12 hours or more postprandially [44]. Despite all blood draws taken after a minimum 10-hour fast, recent seafood intake may have contributed to the TMAO concentration pool, and therefore, incorporating a standardised pre-test diet as a control should be a consideration for future studies. Furthermore, the increased fibre intake in this study may have confounded the putative negative effect of protein due to the modest but well-known consequences of dietary fibre on reducing cholesterol and body weight [66]. However, it should be noted that the diet was designed within the guidelines of healthy eating and so within the context of a healthy diet, the results are relevant. Finally, this study was conducted with healthy elderly men, therefore, the findings may not extend to healthy women in the same age bracket. Given the known gender differences in CVD risk biomarkers, particularly post menopause [67], further investigation in women is required. 

## 5. Conclusions

In this study, a diet containing 1.6 g protein/kg bodyweight/day increased circulating TMAO concentrations. However, there were no changes in other traditional measures of CVD risk, suggesting that TMAO was regulated by protein intake in elderly males. Based on the variable response, it is also likely that TMAO production and clearance is contingent on individualistic and multi-faceted responses drawn from dietary protein load, gut microbiota composition, and kidney function. The elevated microbial product TMAO seen in this study further demonstrates the potential impact of the microbiome in metabolic and cardiovascular health, where further investigation on the composition and activity in the gut microbiota to detect correlations is warranted.

## Figures and Tables

**Figure 1 nutrients-11-02207-f001:**
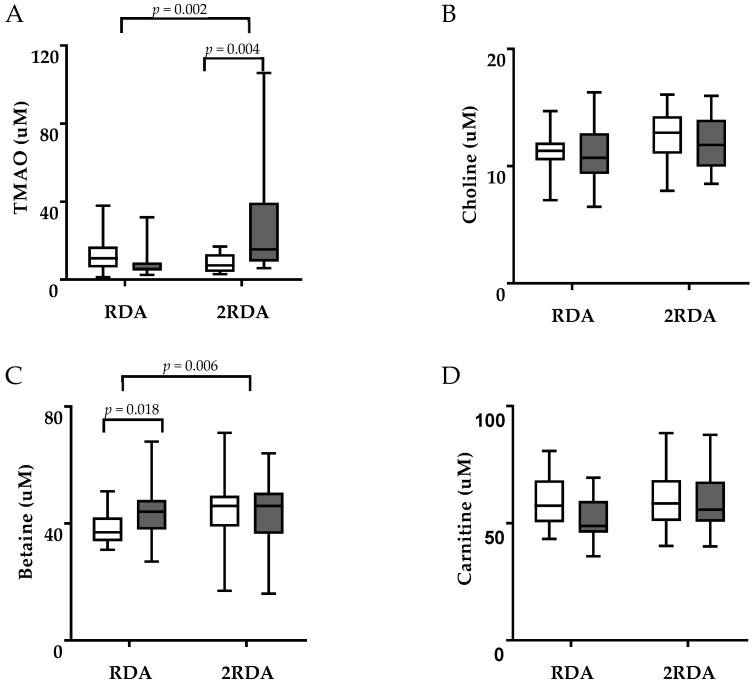
Box-whisker plots illustrating concentrations of (**A**) TMAO, (**B**) choline, (**C**) betaine, (**D**) carnitine in plasma of the RDA and 2RDA groups pre-intervention (uncoloured boxes) and after 10 weeks (coloured boxes). The central lines represent means and the error bars represent minimum and maximum values. TMAO: trimethylamine-N-oxide. Two-way repeated measures ANOVA was used for comparison and the difference between time points in same group was controlled using the Sidak post hoc procedure.

**Figure 2 nutrients-11-02207-f002:**
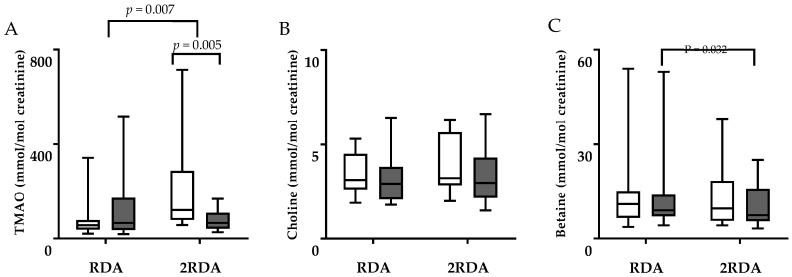
Box whisker plots illustrating concentrations of (**A**) TMAO, (**B**) choline, and (**C**) betaine in urine of the RDA and 2RDA groups pre-intervention (uncoloured boxes) and after 10 weeks (coloured boxes). The central lines represent means and the error bars represent minimum and maximum values. TMAO: trimethylamine-N-oxide. Two-way repeated measures ANOVA was used for comparison and difference between time points in same group was controlled using the Sidak post hoc procedure.

**Figure 3 nutrients-11-02207-f003:**
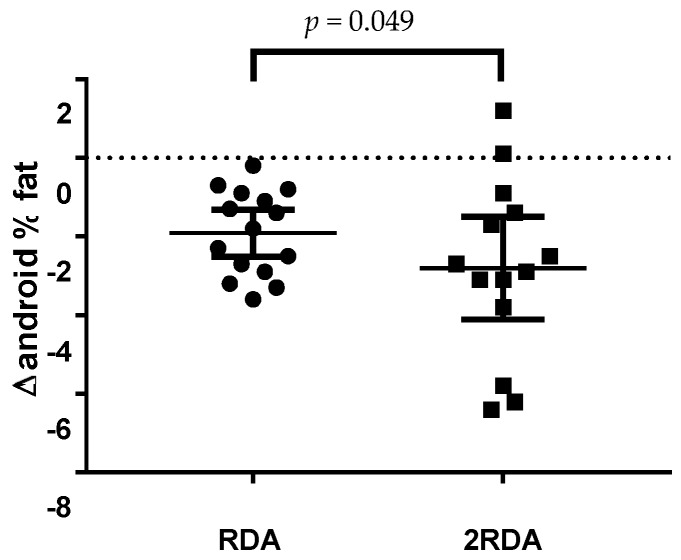
Change in android % fat from before to after 10 weeks of consuming the RDA (circles) and 2RDA (squares) diets. Each point represents a participant. The central lines represent means and the error bars represent SD. Δ, absolute change; Two-way repeated measures ANOVA was used for comparison.

**Table 1 nutrients-11-02207-t001:** Participant characteristics pre-intervention.

Parameter	RDA Group	2RDA Group
	Mean ± SD or Number	Mean ± SD or Number
Count ^1^	15	14
Age (years)	74.7 ± 3.9	73.7 ± 3.3
Height (cm)	172.8 ± 8.2	171.7 ± 5.5
Weight (kg)	85.7 ± 20.5	83.0 ± 8.3
BMI (kg/m^2^)	28.4 ± 5.1	28.2 ± 3.3
Medication usage (n)		
Statin	4	6
ACE inhibitor	4	3
Aspirin	4	2
Calcium channel blocker	1	2
Proton pump inhibitors	0	2
Alpha blocker	1	2
Beta blocker	1	2
Xanthine oxidase inhibitor	1	1

BMI—body mass index, ACE—angiotensin converting enzyme. ^1^ Only the participants who completed the study are included (n = 29).

**Table 2 nutrients-11-02207-t002:** Servings per day of egg, fish, red meat, white meat, dairy intake, and fruit and vegetables for the RDA and 2RDA groups pre- and post-intervention.

Food Group	RDA (Mean ± SD)		2RDA (Mean ± SD)		Effect (*p*) ^a^		
	Pre	Post	Pre	Post	Time	Diet	Time × Diet
Egg ^1^	0.7 ± 0.6	0.6 ± 0.4	0.7 ± 0.5	1. ± 0.4 ^† #^	0.047 *	0.054	0.021 *
Fish ^2^	0.4 ± 0.4	0.1 ± 0.1 ^#^	0.4 ± 0.5	0.6 ± 0.3 ^†^	0.489	0.013 *	0.002 *
Red meat ^2^	1.2 ± 0.9	0.2 ± 0.1 ^#^	1.3 ± 0.7	1.1 ± 0.3 ^†^	<0.001 *	0.003 *	0.011 *
White meat ^2^	0.4 ± 0.5	0.2 ± 0.1	0.3 ± 0.4	0.9 ± 0.1 ^† #^	0.024 *	0.001 *	<0.001 *
Dairy ^3^	1.1 ± 0.8	0.8 ± 0.5	1.1 ± 0.8	2.4 ± 0.7 ^† #^	0.014 *	<0.001 *	<0.001 *
Fruit & vegetables ^4^	5.3 ± 3	8.5 ± 1.6 ^#^	4.4 ± 2.1	8.6 ± 2.1 ^#^	<0.001 *	0.41	0.535

Serving size calculated as ^1^ one egg; ^2^ 100 g cooked weight; ^3^ 250 mL milk or 150 g yoghurt, or 50 g cheese; ^4^ one cup. Fish includes fatty fish, white fish, and shellfish; red meat includes beef, lamb, venison, and pork; white meat includes poultry; dairy includes milk, yoghurt, cheese, and ice-cream. * Significant main effect or interaction, *p* < 0.05. # Different from pre-intervention within the same group, *p* < 0.05. † Different between diets at indicated time point. *p* values were controlled using the Sidak post hoc procedure. ^a^ Main effects and interactions were calculated by two-way repeated measures ANOVA.

**Table 3 nutrients-11-02207-t003:** Fasting measurements of circulatory cholesterol, triglycerides, glucose, insulin, CRP and anthropometric measurements of the RDA and 2RDA groups pre- and post-intervention.

Variable	RDA (Mean ± SD)		2RDA (Mean ± SD)		Effect (*p*) ^a^		
	Pre	Post	Pre	Post	Time	Diet	Time × Diet
Lipids & lipoproteins							
T-C (mmol/L)	4.7 ± 0.9	4.5 ± 1.2	4.6 ± 0.9	4.8 ± 1.2	0.82	0.877	0.055
HDL-C (mmol/L)	1.4 ± 0.5	1.4 ± 0.6	1.3 ± 0.3	1.3 ± 0.3	0.89	0.559	0.662
LDL-C (mmol/L)	3.0 ± 1	2.8 ± 1.1	3.0 ± 0.9	3.2 ± 1.3	0.7	0.668	0.049 *
TAG (mmol/L)	1.2 ± 0.4	1.2 ± 0.4	1.2 ± 0.8	1.1 ± 0.5	0.55	0.631	0.871
Metabolic & inflammatory biomarkers							
Glucose (mmol/L)	5.7 ± 0.5	5.6 ± 0.5	5.7 ± 0.6	5.5 ± 0.4	0.020 *	0.639	0.358
Insulin (µU/mL)	9.7 ± 3.9	9.6 ± 3.7	12.5 ± 10.7	9.9 ± 5.6	0.391	0.421	0.42
HbA1c (mmol/mol)	36.1 ± 5	36.8 ± 4.4	36.9 ± 3.7	36.9 ± 3.3	0.397	0.761	0.305
HOMA-IR	2.5 ± 1.2	2.4 ± 1.1	3.5 ± 3.5	2.4 ± 1.3	0.305	0.439	0.425
Matsuda Index	3.2 ± 1.7	3.2 ± 1.8	3.2 ± 1.9	3.5 ± 2.1	0.15	0.649	0.216
CRP (mg/L)	1.8 ± 2.1	2.1 ± 3.3	2 ± 3.1	2.3 ± 1.6	0.644	0.862	0.967
Anthropometrics							
Total body fat (kg)	25.4 ± 11.5	23.9 ± 11.0 ^#^	23.5 ± 6.8	21.8 ± 6.8 ^#^	<0.001 *	0.534	0.954
Android fat (%)	38.4 ± 11	36.7 ± 11 ^#^	39.9 ± 8	36.9 ± 8.0 ^#^	<0.001 *	0.831	0.049 *
WC (cm)	104.8 ± 15	99.6 ± 14.1 ^#^	101.5 ± 7.6	96 ± 7.5 ^#^	<0.001 *	0.377	0.765
Systolic BP (mmHg)	143 ± 14	144 ± 11	142 ± 18	141 ± 16	0.839	0.647	0.663
Diastolic BP (mmHg)	75 ± 7	74 ± 7	76 ± 9	75 ± 9	0.126	0.668	0.975

T-C, total cholesterol; HDL-C, high density lipoprotein cholesterol; LDL-C, low density lipoprotein cholesterol; TAG, triacylglycerides; HbA1c, glycated haemoglobin; HOMA-IR, homeostatic model assessment of insulin resistance; CRP, C-reactive protein; WC, waist circumference; BP, blood pressure. Insulin and glucose concentrations during the OGTT were used to calculate the Matsuda Index. * Significant main effect or interaction, *p* < 0.05. # Different from pre-intervention within the same group, *p* < 0.05. † Different between diets at indicated time point. *p* values were controlled using the Sidak post hoc procedure. ^a^ The main effects and interactions were calculated by two-way repeated measures ANOVA.

**Table 4 nutrients-11-02207-t004:** Measurements of creatinine from plasma and urine, calculated creatinine clearance, and calculated fractional renal excretion of TMAO of the RDA and 2RDA groups pre- and post-intervention.

Variable	RDA (Mean ± SD)		2RDA (Mean ± SD)		Effect (*p*) ^a^		
	Pre	Post	Pre	Post	Time	Diet	Time × Diet
**Renal biomarkers**							
Plasma Cr (µmol/L)	90.8 ± 18.9	82 ± 16.1 ^#^	98.1 ± 11	97.4 ± 16.7 ^†^	0.010 *	0.057	0.027 *
Urinary Cr (mmol/L)	6.3 ± 2.9	5.7 ± 3.8	6.8 ± 2.1	6.3 ± 2.3	0.174	0.673	0.730
Cr clearance (mL/min)	128.1 ± 50.5	101 ± 37.1 ^#^	116.8 ± 25.2	116 ± 24.4	0.010 *	0.882	0.013 *
Fractional renal excretion of TMAO (%)	76.7 ± 68.7	173.2 ± 213.7	305.2 ± 288.3 ^†^	54.3 ± 49.4 ^† #^	0.095	0.619	0.002 *

Cr, creatinine; TMAO, trimethylamine-N-oxide. * Significant main effect or interaction, *p* < 0.05. # Different from pre-intervention within the same group, *p* < 0.05. † Different between diets at indicated time point. *p* values were controlled using the Sidak post hoc procedure. ^a^ Main effects and interactions were calculated by two-way repeated measures ANOVA.

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
