# Peer review of "Protein Intake at Twice the RDA in Older Men Increases Circulatory Concentrations of the Microbiome Metabolite Trimethylamine-N-Oxide (TMAO)"

_nutrients, 2019, doi:10.3390/nu11092207_

Round 1
Reviewer 1 Report
This study shows the relationship between an intake of excess protein and an accumulation of TMAO, trimethylamine-N-oxide, in healthy elder men. This is interesting. However, women and young people should have participated in this study, as the authors have mentioned it. In addition, the reviewer thought that the determining TMAO were required in cardiovascular disease (CVD) patients, not in healthy subjects, if it could be used as an alternative to traditional biomarkers of CVD. This is the weakest point in this study. Moreover, discussion seems to be too long. L364-373 indicates the relationship between TMAO and CVD. However, this study did not examine such relation. Therefore the reviewer did not think that this part was needed. And, the discussion includes a mix of results of animal studies and clinical studies. Maybe reports of clinical studies are more suitable in this study.
Reviewer 2 Report
Here authors present this study to determine the effect of consumption of dietary protein at twice the current recommended dietary allowance (RDA) compared to consumption at the RDA on Trimethylamine-N-oxide (TMAO) concentrations and traditional biomarkers of cardiovascular and metabolic disease risk in the elderly.Their results indicated that the 2RDA diet increased circulatory TMAO, but unexpectedly decreased renal excretion of TMAO, and increased the LDL cholesterol level. However, the 2RDA diet did not affect other biomarkers of CVD risk and insulin sensitivity. Authors concluded that circulatory TMAO is responsive to changes in dietary protein intake.
Here is my specific comments.
Major comments:
What is the criteria of inclusion and exclusion for participants used in this study? Why did authors not choose women? Why did authors use the twice the RDA? Does it mean the high dietary protein intake?Author Response
Please see the attachment
Round 2
Reviewer 1 Report
The TMAO levels are required in cardiovascular disease (CVD) patients after all that, if the authors expect that it could be used as the marker for a diagnosis of a risk of CVD.
However, the reviewer appreciates the response and efforts which have been done by the authors to enhance the quality of the work. Moreover, the reviewer understands that a high protein diet has been linked to an increase in TMAO level in the plasma of the healthy elder men. Therefore, it is suggested that the authors change the title (delete the word ‘CVD’) and focus on the point which a high protein diet could elevate a TMAO level, rather than expecting TMAO to become the biomarker, in the text (abstract, introduction, discussion and conclusion).
Reviewer 2 Report
In the revised version,authors have addressed the major points of criticism raised by the reviewers.The paper has been improved.
